# Safety Evaluation of an Alpha-Emitter Bismuth-213 Labeled Antibody to (1→3)-β-Glucan in Healthy Dogs as a Prelude for a Trial in Companion Dogs with Invasive Fungal Infections

**DOI:** 10.3390/molecules25163604

**Published:** 2020-08-08

**Authors:** Muath Helal, Kevin J. H. Allen, Hilary Burgess, Rubin Jiao, Mackenzie E. Malo, Matthew Hutcheson, Ekaterina Dadachova, Elisabeth Snead

**Affiliations:** 1College of Pharmacy and Nutrition, University of Saskatchewan, Saskatoon, SK S7N 5E5, Canada; muh166@mail.usask.ca (M.H.); kja782@mail.usask.ca (K.J.H.A.); ruj501@mail.usask.ca (R.J.); mem510@mail.usask.ca (M.E.M.); 2Western College of Veterinary Medicine, Saskatoon, SK S7N 5B4, Canada; hilary.burgess@usask.ca (H.B.); ecs212@mail.usask.ca (E.S.); 3Safety Resources, University of Saskatchewan, Saskatoon, SK S7N 5C5, Canada; matt.hutcheson@usask.ca

**Keywords:** radioimmunotherapy, bismuth-213, invasive fungal infections, 1-3-beta-glucan, dogs

## Abstract

**Background**: With the limited options available for therapy to treat invasive fungal infections (IFI), radioimmunotherapy (RIT) can potentially offer an effective alternative treatment. Microorganism-specific monoclonal antibodies have shown promising results in the experimental treatment of fungal, bacterial, and viral infections, including our recent and encouraging results from treating mice infected with *Blastomyces dermatitidis* with ^213^Bi-labeled antibody 400-2 to (1→3)-β-glucan. In this work, we performed a safety study of ^213^Bi-400-2 antibody in healthy dogs as a prelude for a clinical trial in companion dogs with acquired invasive fungal infections and later on in human patients with IFI. **Methods**: Three female beagle dogs (≈6.1 kg body weight) were treated intravenously with 155.3, 142.5, or 133.2 MBq of ^213^Bi-400-2 given as three subfractions over an 8 h period. RBC, WBC, platelet, and blood serum biochemistry parameters were measured periodically for 6 months post injection. **Results**: No significant acute or long-term side effects were observed after RIT injections; only a few parameters were mildly and transiently outside reference change value limits, and a transient atypical morphology was observed in the circulating lymphocyte population of two dogs. **Conclusions**: These results demonstrate the safety of systemic ^213^Bi-400-2 administration in dogs and provide encouragement to pursue evaluation of RIT of IFI in companion dogs.

## 1. Introduction

For immunosuppressed patients, such as those suffering from cancer or are post organ transplant, invasive fungal infections (IFI) can be devastating. As causes of both morbidity and mortality, cases of IFI have more than tripled since 1979, highlighting the need for treatment [1]. IFI can also have devastating effects on dogs and cats [2]. New effective treatments for IFI are needed for human and veterinary medicine to overcome resistance issues, issues associated with poor bioavailability, and the high cost of oral antifungals in dogs. Radioimmunotherapy (RIT) uses antigen–antibody interaction to deliver sufficient activities of ionizing radiation to cells to induce DNA double-strand breaks and alter cell membrane and intracellular components for cell apoptosis while preserving healthy tissues. This treatment has demonstrated efficacy in primarily nonsolid tumors such as non-Hodgkin lymphomas [3]. RIT offers several advantages over other therapeutics: (1) Due to the nature of ionizing radiation, RIT physically destroys the desired cells, and does not just abolish a single pathway; (2) the total cell destruction makes it very difficult for a drug resistance mechanism to develop; (3) RIT does not rely on a healthy immune system to be effective, making the immune status of the patient less important; (4) the specific nature of RIT means there are less off-target effects, which results in lower toxicity compared to that of conventional chemotherapies. However, RIT is still subject to radioresistance phenomena, especially for big solid tumors, and induces hematoxicity in some patients. Additionally, we were the first to show that microorganism-specific monoclonal antibodies can be a viable therapy for viral, bacterial, and fungal infections in an experimental environment ([4,5]). RIT for infections potentially offers higher specificity and lower toxicity than cancer RIT as it targets microbial antigens which have no or very little homology with human proteins. In contrast to cancer, where every cancerous cell must be destroyed either directly by crossfire and bystander effects or indirectly via abscopal effect to prevent the resurgence of the disease, low numbers of microbial cells remaining post RIT can be eliminated by the immune system. Previously, we theorized that beta-glucan, a surface-expressed antigen that is shared by many major IFI-causing pathogens, would make an ideal target for RIT [6]. To show this, we utilized *Cryptococcus neoformans* and *Candida albicans*, as they are highly present. The in vitro results demonstrated that when labeled with the alpha-particle-emitting radionuclide Bismuth-213 (^213^Bi) the antibodies to the pan-antigens killed a very high percentage (80–100%) of the fungal cells [7]. ^213^Bi, an alpha-emitter with a short physical half-life of 46 min, delivers its radiation impact within a short period of time, which can match the doubling time of the majority of the fungal pathogens. It gives ^213^Bi an advantage over other longer-lived alpha emitters such as Actinium-225. Next, we were able to show that this approach can be translated in vivo by infecting mice with *Blastomyces dermatitidis* and targeting the pan-antigen beta-glucan [8]. *B. dermatitidis* was chosen as an IFI model as it is endemic to parts of Canada and causes devastating infections in immunocompetent people, companion dogs, and immunocompromised patients [2]. In this work, we performed a safety study of ^213^Bi-400-2 antibody in healthy dogs as a prelude for a clinical trial in companion dogs with invasive fungal infections.

## 2. Results

On separate days, three dogs were intravenously administered 155.3, 142.5, or 133.2 MBq (1→3)-β-glucan-targeting antibody 400-2 labeled with ^213^Bi, respectively. Figure 1 shows the values for red blood cell (RBC), white blood cell (WBC), and platelet counts throughout the study. These parameters are of particular importance for radiolabeled antibody-based treatment as they reflect possible reversible or nonreversible toxicity to the bone marrow. In all three dogs, these parameters were within laboratory reference intervals at all time points and generally within the reference change value (RCV) limits throughout the duration of the study.

In two of the patients (dogs 1 and 2), atypical cells, most consistent with cells of lymphoid origin, were noted on days 7 and 14, respectively (Figure 2). These cells were intermediate to large in size and round to mildly irregular in shape. Cytoplasm was of low to moderate volume, basophilic, and occasionally had few pinpoint clear vacuoles or pale areas. Nuclei were round, but in rare cells could appear binucleate or possibly lobular, with ropey, occasionally clumped, chromatin (Figure 2).

These atypical cells were not a persistent finding, and they were not discernible in any of the follow-up complete blood counts (CBCs) in the two affected dogs. The mild decrease in lymphocyte numbers could reflect a transient elevation in cortisol (stress); however, given the brief appearance of atypical cells, a very transient influence of the treatment on lymphoid populations in the bone marrow and other organs leading to lympholysis cannot be entirely ruled out. The changes in monocyte numbers did not follow a consistent pattern between subjects, and while decreased numbers are typically considered clinically insignificant, the mild elevations could reflect an increased demand for tissue macrophages. Although within the RCV limits, mean corpuscular volume (MCV) showed a consistent downward trend from day 3 to day 14 in all three dogs, and continued to 30 days in dogs 1 and 3 (Figure 3). While there was an upward trend in all dogs by day 90, the values again decreased at 180 days. This finding can be associated with iron-limited erythropoiesis, although its connection to the test therapy is unclear. The dogs were all on a low-protein, plant-based diet which may have influenced iron stores independent of the test therapy.

A selection of serum biochemical analytes are shown in Figure 4, Figure 5, Figure 6 and Figure 7. With the exception of a temporary decrease in creatinine at day 90 for dog 3, and mild elevations in urea relative to the RCV at day 180 in dogs 1 and 2, urea and creatinine were within RCV throughout the study (Figure 4).

Chloride, phosphorus, and potassium were concurrently elevated relative to the RCV in dog 2, and potassium was elevated relative to the RCV in dog 1, suggesting altered hydration status was the likely cause of the urea elevation in both dogs (Figure 5). Hydration status was also the likely cause of elevations in potassium at day 30 in dog 2 and day 90 in dogs 1 and 3, as again sodium showed a similar increase or was above the RCV at these same time points and was accompanied by a chloride level above the RCV in dog 3 at day 90. There was a slight decrease in potassium relative to the RCV in dogs 1 and 3 at day 30 and dog 2 on day 90, which may have been related to decreased intake. Despite these changes, relative to the RCV, potassium was within laboratory reference intervals throughout the study.

With the exception of alkaline phosphatase (ALP) in dog 2, which significantly increased above RCV at day 30 but corrected by day 90, and slight increases in alanine aminotransferase ALT at day 180 in dogs 2 and 3, all hepatobiliary enzymes remained within the RCV limits throughout the study (Figure 6).

The bilirubin and albumin were somewhat low, relative to laboratory reference intervals, in all three dogs at baseline and throughout the 6-month study (Figure 7). Total bilirubin was within RCV limits for all dogs throughout the study. Albumin was decreased relative to the RCV in dog 1 (at 90 days), dog 2 (at 30 days), and dog 3 (at 90 and 180 days). In dogs 1 and 2, these were transient changes. However, in dog 3, who had the lowest albumin at the start of the study, there was suspicion of a pre-existing protein-losing enteropathy (PLE). The albumin results were consistently and significantly lower than laboratory reference intervals at baseline and throughout the study in dog 3. Suspicion of PLE was further supported by the development of ascites, characterized as a pure transudate, at day 90 which was responsive to a hypoallergenic diet and prednisolone therapy.

Cholesterol was decreased concurrently, relative to RCV, in dog 1 at 90 days and in dog 3 at 180 days (Figure 8). There was a mild increase in cholesterol in dog 2 at 30 days, with a concurrent mild elevation of ALP relative to the RCV. This likely reflected a transient cholestasis, although insufficient to influence bilirubin concentrations. In the same dog, cholesterol had decreased relative to the RCV at 90 days and showed a slight elevation above the RCV at 180 days.

ALP corrected by 90 days. Elevations in cholesterol, relative to the RCV, were also noted in dog 3 at 30 and 90 days. These elevations were transient and nonspecific. Glucose was found to decrease relative to the RCV in dog 1 on days 90 and 180 (Figure 8). Delayed serum separation was thought to be the most likely explanation for this change. Mild elevations in glucose relative to the RCV were noted in dog 2 at day 90 and dog 3 at days 30 and 90, which could be explained by a cortisol response, further supported in dog 3 at 30 days by a concurrent decrease in lymphocyte numbers. However, in dog 3, the baseline glucose was low relative to the laboratory reference intervals (likely storage artifact). As the RCV was based on this artifactually low value, any future sample not influenced by storage artifact would be expected to exceed the upper limit of the RCV. CK was elevated above the RCV, but remained within laboratory reference intervals, at day 30 in dog 1, likely reflecting mild myocyte leakage related to restraint during sample collection.

## 3. Discussion

Overall, throughout the study, changes outside the RCV were few, mild, and transient, attesting to the absence of systemic toxicity of ^213^Bi-400-2 antibody. In regard to the radiation safety, the research and veterinary staff administering the ^213^Bi-labeled antibody and providing animal care received a combined dose of no more than 6 µSv personal dose equivalent at the depth of 10 mm (Hp(10)) per handled dog as measured by electronic dosimeters due to the 440 keV gamma emissions from the ^213^Bi. The staff exposures indicate that, with proper precautions, personnel exposure will not be limiting in a clinical trial setting. The short half-life of ^213^Bi makes the required isolation period less burdensome than using longer-lived radionuclides.

*B. dermatitidis* has relatively high prevalence in different areas in Canada and the United States [2,9], and the endemic regions may be increasing, as evidenced by reports in New York, Vermont, Texas, Nebraska, and Kansas [10], which makes it a relevant fungal pathogen to be treated with radiolabeled antibodies to a pan-antigen beta-glucan. Treatment of mice infected intratracheally with *B. dermatitidis* by ^213^Bi-400-2 antibody given as a single 5.6 MBq (277.5 MBq/kg body weight) intraperitoneal injection reduced the fungal burden in the lungs by two logs [8]. For treatment of the dogs in this study, approximately 18.5 MBq ^213^Bi-400-2 antibody per kg body weight was administered, reflecting the difference in body weight to body surface area ratios between mice and dogs. In this regard, the studies of T-cell ablation in dogs with ^213^Bi-labeled anti-TCRαβ antibody showed that the injected activities in the range of 137 to 207 MBq/kg did not cause normal organ damage [11,12]. In addition, because (1→3)-β-glucan is a fungal antigen with no homology to human proteins, the antibodies to (1→3)-β-glucan are expected to have very low cross-reactivity with human and canine cells, thus decreasing the potential side effects from off-target antibody mediated injury associated with radiolabeled antibody administration. ^213^Bi-labeled whole antibodies have been successfully used in treatment of patients with acute myeloid leukemia and melanoma [3]. Currently, there is a worldwide effort to increase the availability of ^213^Bi, which will enable the expansion of ^213^Bi-labeled whole antibodies’ use for a variety of oncological and infectious disease applications.

## 4. Materials and Methods

### 4.1. Radionuclides, Radiolabeling, and Quality Control of the Radiolabeled Antibody

The (1→3)-β-glucan-targeting 400-2 antibody (Biosupplies Australia Pty. Ltd., Parkville, Australia) was first conjugated to the chelating agent *C*-functionalized *trans*-cyclohexyldiethylene-triamine penta-acetic acid derivative (CHXA″) (Macrocyclics, San Antonio, TX, USA). Conjugation and labeling were carried out according to a previously published method [13]. To summarize, conjugation was carried out at 37 °C for 1.5 h in a sodium carbonate buffer at pH 8.5 that had been previously run through a chelex column to remove any trace metal impurities. After the conjugation, the reaction was exchanged into a chelexed 0.15 M ammonium acetate buffer at pH 6.5. The ^213^Bi/^225^Ac radionuclide generator was purchased from Oak Ridge National Laboratory (Oak Ridge, TN, USA). ^213^Bi was eluted from the ^213^Bi/^225^Ac radionuclide generator using 300 µL 0.1 M hydroiodic acid (HI) solution followed by 300 µL milliQ H_2_O [14]. The elution pH was adjusted to 6.5 with 80 µL of 5 M ammonium acetate buffer (chelexed) prior to the addition of the desired amount of CHXA’’ conjugated antibody (370:1 kBq/µg specific activity, 55.5 kBq/pmol) and the reaction mixture was heated for 5 min at 37 °C with shaking, followed by an addition of 3 µL of 0.05 M EDTA solution. The ^213^Bi-labeled antibody was then purified on a 0.5 mL Amicon disposable size exclusion filter (30K MW cut-off, Fisher, Ottawa, ON, Canada. The percentage of radiolabeling was measured by silica gel instant thin-layer chromatography (SG-iTLC, Agilent Technologies, Santa Clara, CA, USA). using 0.15 M ammonium acetate buffer as the eluent. SG-iTLCs were cut in half, and each half was read on a Wizard2 2470 Automatic Gamma Counter (R_f_ = 0 containing radiolabeled antibody, R_f_ = 1 containing free ^213^Bi (Perkin Elmer, Waltham, MA, USA). Only the batches with >98% radiochemical purity were used for administration to the dogs.

### 4.2. Administration of Radiolabeled Antibody to Dogs and Follow up Analyses

The study design was approved by the University of Saskatchewan’s Animal Research Ethics Board and adhered to the Canadian Council on Animal Care guidelines for humane animal use (Animal Use Protocol 20190091, approved on August 7, 2019). Three 1.5 years old, spayed female beagle dogs (5.91–6.1 kg body weight) who were deemed healthy and suitable for radiolabeled antibody administration were utilized. Dogs were allowed to acclimate for one week prior to use, and their appetite and activity levels were monitored before the initiation of the administration of radiolabeled antibody. All dogs were eating a homemade, balanced, low-protein diet that met all minimum Association of American Feed Control Officials (AFFCO) standards. On the injection day, vital parameters such as normal body temperature, body weight, and heart rate were recorded and monitored. Ethylenediamine tetra-acetic acid (EDTA)-anticoagulated blood, serum, and urine were submitted to Prairie Diagnostic Services (PDS; Saskatoon, SK, Canada) for baseline complete blood count (CBC) (Advia Hematology Analyzer, Siemens Healthcare, Germany), serum biochemistry (Cobas 311, Hitachi High-Technologies Corporation, Tokyo, Japan), and urinalysis, respectively. On separate days, three dogs were intravenously administered 155.3, 142.5, or 133.2 MBq (1→3)-β-glucan-targeting antibody 400-2 labeled with ^213^Bi (specific activity 370 kBq/µg), respectively. The injected activity was divided into three subfractions separated by 2–3 h for each dog. The dogs were closely monitored between the injections and after the last subfraction for 12–14 h. EDTA-anticoagulated blood from each dog was submitted to PDS 3, 7, 14, 30, 90, and 180 days post injection for CBC monitoring. Serum collected on 30, 90, and 180 days post injection was used to measure serum biochemistry levels. Urine samples were collected by ultrasound-guided cystocentesis on days 0, 90, and 180. CBC evaluation included total white blood cell count (WBC), with manual differential leukocyte count, red blood cell count (RBC), hemoglobin concentration (Hgb), hematocrit (Hct), mean corpuscular volume (MCV), mean corpuscular hemoglobin concentration (MCHC), mean corpuscular hemoglobin (MCH), platelet count, and evaluation of red blood cell morphology. Serum biochemical analytes evaluated were sodium, potassium, chloride, calcium, phosphorus, magnesium, urea, creatinine, lipase, glucose, cholesterol, total bilirubin, alkaline phosphatase (ALP), gamma glutamyltransferase (GGT), alanine aminotransferase (ALT), creatinine kinase (CK), total protein, albumin, and calculated globulin. Upper and lower limits for individual variation were calculated from the baseline value for each analyte using canine CV_I_ data from VetBiologicalVariation.org. For those analytes where one or more values were outside of these limits, a reference change value was calculated using RCV (95%) data from VetBiologicalVariation.org.

### 4.3. Radiation Safety for the Research and Veterinary Staff

Research staff administering the ^213^Bi-labeled antibody and providing animal care wore electronic dosimeters (DMC 3000, Mirion Technologies, Smyrna, GA, USA). Animals were housed in an isolation area suitable for work with radioisotopes for approximately 8 h after the final administration and released from isolation when less than 0.1% of the administered ^213^Bi remained.

## 5. Conclusions

In conclusion, our results have demonstrated an acceptable safety profile of (1→3)-β-glucan-targeting ^213^Bi-400-2 antibody in dogs. RIT may be a useful adjunct therapy for invasive fungal infections and have a role to play in speeding resolution of an infection by helping the body to reduce the infectious load to a lower level where either the immune system or other therapies may be more effective. The clinical trial will recruit canine patients with invasive fungal infections and will also provide the information needed for clinical translation into human patients with IFI.

## Figures and Tables

**Figure 1 molecules-25-03604-f001:**
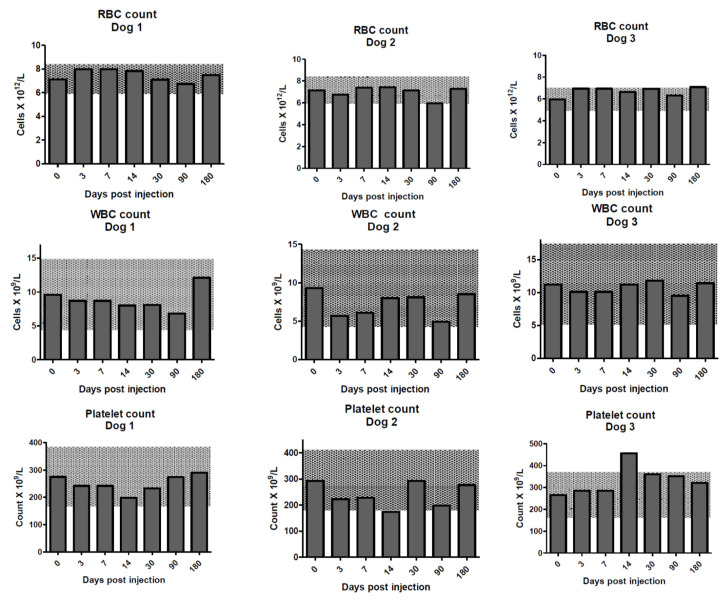
Red blood cell (RBC) counts (**upper row**), white blood cell (WBC) counts (**middle row**), and platelet counts (**lower row**) for the dogs on days 0–180 post treatment with ^213^Bi-400-2 antibody. Shaded areas show reference change values (RCVs).

**Figure 2 molecules-25-03604-f002:**
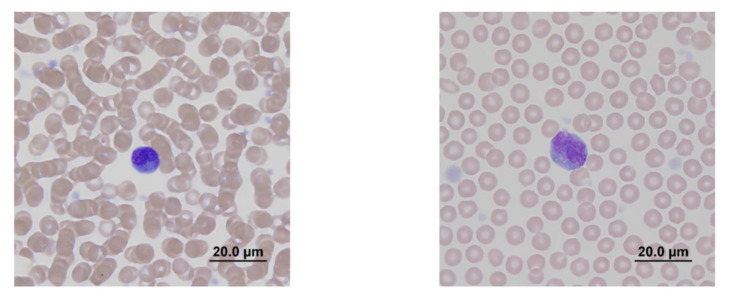
Direct smears of peripheral blood from dogs 1 (**left micrograph**) and 2 (**right micrograph**) showing examples of atypical cells, most consistent with lymphoid origin. These cells are intermediate to large in size with low volumes of basophilic cytoplasm, occasionally containing few small clear vacuoles. Nuclei in these two cells are cleaved to binucleate or lobular with clumped chromatin. Modified Wright stain. Bar = 20.0 µm.

**Figure 3 molecules-25-03604-f003:**
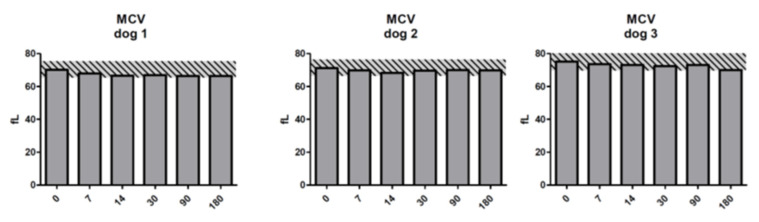
Mean corpuscular volume (MCV) for the dogs on days 0–180 post treatment with ^213^Bi-400-2 antibody. Shaded areas show reference change values (RCVs). X axis shows days post injection of radiolabeled antibody.

**Figure 4 molecules-25-03604-f004:**
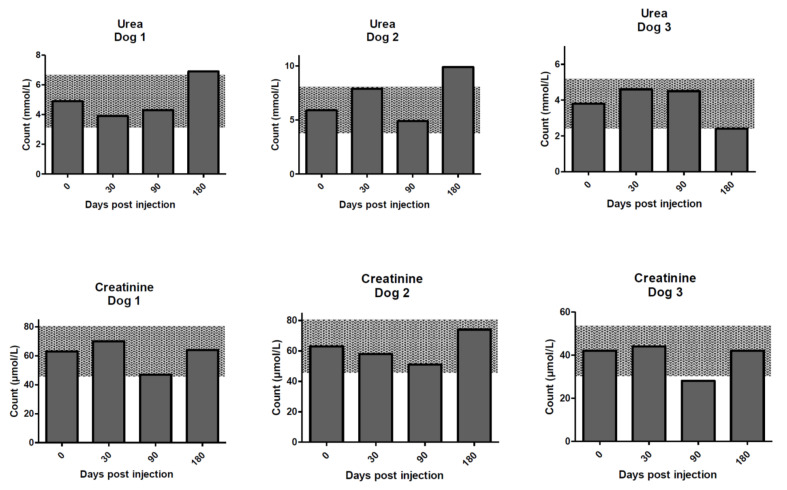
Renal analytes for the dogs on days 0–180 post treatment with ^213^Bi-400-2 antibody. **Upper row**—urea; **lower row**—creatinine. Shaded areas show reference change values (RCVs).

**Figure 5 molecules-25-03604-f005:**
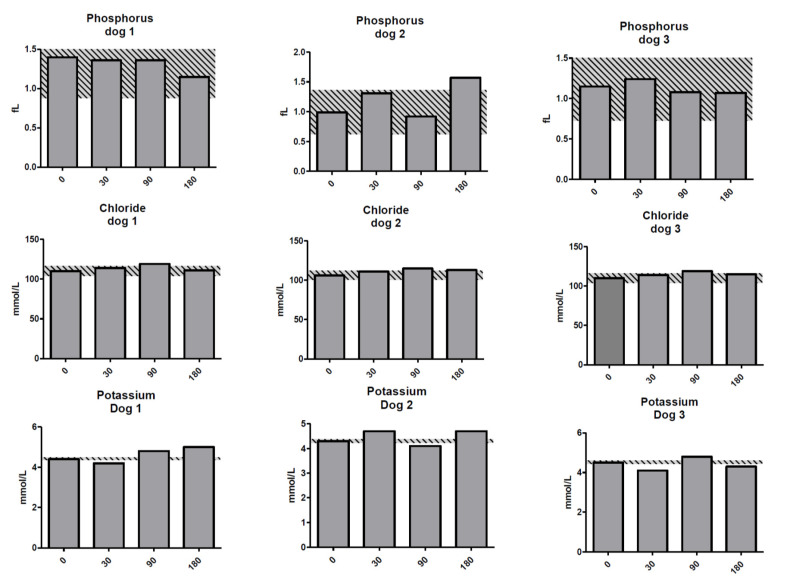
Phosphorus, chloride, and potassium for the dogs on days 0–180 post treatment with ^213^Bi-400-2 antibody. **Upper row**—phosphorous; **middle row**—chloride; **lower row**—potassium. Shaded areas show reference change values (RCVs). X axis shows days post injection of radiolabeled antibody.

**Figure 6 molecules-25-03604-f006:**
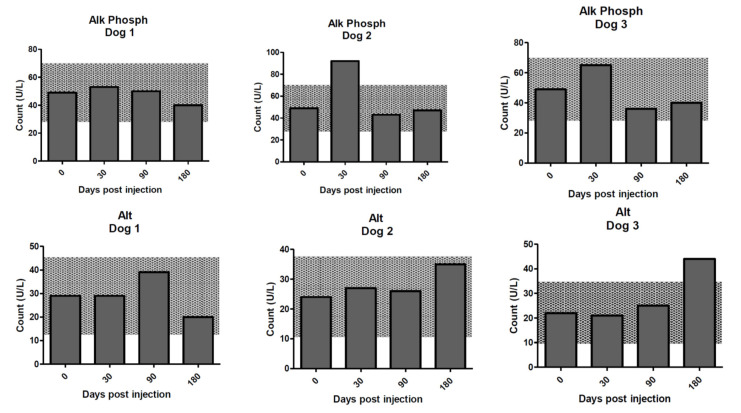
Liver enzyme measurements for the dogs on days 0–180 post treatment with ^213^Bi-400-2 antibody. **Upper row**—alkaline phosphatase; **lower row**—alanine aminotransferase. Shaded areas show reference change values (RCVs).

**Figure 7 molecules-25-03604-f007:**
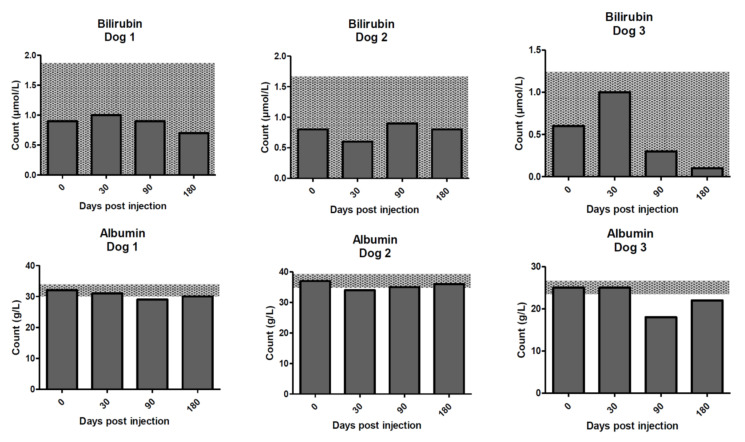
Bilirubin and albumin concentrations for the dogs on days 0–180 post treatment with ^213^Bi-400-2 antibody. **Upper row**—bilirubin; **lower row**—albumin. Shaded areas show reference change values (RCVs).

**Figure 8 molecules-25-03604-f008:**
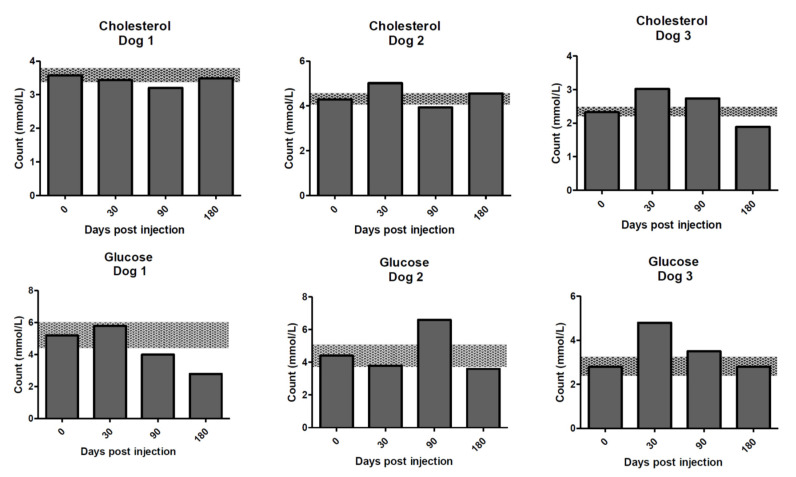
Cholesterol and glucose concentrations for the dogs on days 0–180 post treatment with ^213^Bi-400-2 antibody. **Upper row**—cholesterol; **lower row**—glucose. Shaded areas show reference change values (RCVs).

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
