# Peer review of "Safety Evaluation of an Alpha-Emitter Bismuth-213 Labeled Antibody to (1→3)-β-Glucan in Healthy Dogs as a Prelude for a Trial in Companion Dogs with Invasive Fungal Infections"

_molecules, 2020, doi:10.3390/molecules25163604_

Round 1
Reviewer 1 Report
The article from Helal and co-workers is about the evaluation of the toxicity with Bismuth-213-mAb in healthy dog in order to perform further RIT of fungal infections. While the study is interesting and the experiments well conducted there are several issues to solve before being suitable for publication.
- There are several mistakes or approximations in the introduction section concerning RIT.
- L40 “to deliver lethal doses of ionizing radiation to cells”. This is an approximation. RIT consists to deliver sufficient activities of ionizing radiations to induce DNA double strand breaks (or alters cell membrane and intracellular components) for cell apoptosis while preserving healthy tissue. It is not lethal stricto sensus which imply that it will kill everything.
- L40-41 “has demonstrated efficacy in several types of cancer”. Actually, in oncology RIT is mainly effective on non-solid tumors. In humans, RIT received approval exclusively for non-Hodgkin lymphomas (Zevalin, Bexxar) due to important drawbacks on solid tumors (low clearance of mAbs, low penetration of mAbs in the center of the tumors, radioresistance).
- L43 “it is less subject to drug resistance mechanisms”. No, but it is subject to radioresistance phenomenon, especially for big solid tumors.
- L44 “it has low toxicity in comparison to other forms of chemotherapy”. Not true. RIT induces hematotoxicity which is one of the major issues impeding its use in clinic. Plus, a comparison between RIT and chemotherapy which are two different therapies with specific effects is a nonsense.
- L49 “In contrast to cancer when every cancerous cell has to be destroyed to prevent the resurgence of the disease […]. Not false but in the case of radiotherapy this is an approximation. RIT induces direct effects (towards the binding cells and the surrounding cells thanks to cross-fire and bystander effects) but induces also indirect abscopal effect activating immunity reaction leading to the death of non-irradiated cells.
- The authors did not explain their choice of bismuth-213. One sentence or two in the introduction about this choice and the characteristics of alpha-emitter compared to other radionuclides would have been appreciated.
- The authors should use the international units. In the case of radioactivity Curie (Ci) has been replaced by Becquerel (Bq) since several years now. Same for h (hours) instead of hr.
- The authors often used the term “dose” to refer to the injected activity which is false. Internal dose only refers to dosimetry, i.e. the energy absorbed per unit of mass, expressed in Gray (Gy). The injected activity (counts) is not equal to the dose which can only be determined using dosimetry. The authors should replace the term “dose” by “injected activity” to avoid confusion for the reader.
- Dosimetry study, which is probably the most important parameter concerning toxicity is missing. To conclude about the safety of the protocol and go to therapy experiments, an accurate intern dosimetry study is mandatory for each injected activity.
- Final molar activity (Am) of the radiolabeled mAb is missing in the material & methods section.
Author Response
Reviewer 1
The article from Helal and co-workers is about the evaluation of the toxicity with Bismuth-213-mAb in healthy dog in order to perform further RIT of fungal infections. While the study is interesting and the experiments well conducted there are several issues to solve before being suitable for publication. - Response: We would like to thank the Reviewer for his/her encouraging opinion about our work. Below we list the changes which we have made in the revised manuscript following the Reviewer’s recommendations.
- There are several mistakes or approximations in the introduction section concerning RIT.
- L40 “to deliver lethal doses of ionizing radiation to cells”. This is an approximation. RIT consists to deliver sufficient activities of ionizing radiations to induce DNA double strand breaks (or alters cell membrane and intracellular components) for cell apoptosis while preserving healthy tissue. It is not lethal stricto sensus which imply that it will kill everything. - Response: We have changed this sentence which now reads: “Radioimmunotherapy (RIT) uses antigen-antibody interaction to deliver sufficient activities of ionizing radiation to the cells to induce DNA double strand breaks, alter cell membrane and intracellular components for cell apoptosis while preserving healthy tissues and has demonstrated efficacy in primarily non-solid tumors such a non-Hodgkin lymphomas [3]. “ (page 2, Ln 44-47)
- L40-41 “has demonstrated efficacy in several types of cancer”. Actually, in oncology RIT is mainly effective on non-solid tumors. In humans, RIT received approval exclusively for non-Hodgkin lymphomas (Zevalin, Bexxar) due to important drawbacks on solid tumors (low clearance of mAbs, low penetration of mAbs in the center of the tumors, radioresistance). - Response: We have changed this sentence according to the Reviewer’s recommendation - please see the sentence in the answer to the previous comment. (page 2, LN 44-47)
- L43 “it is less subject to drug resistance mechanisms”. No, but it is subject to radioresistance phenomenon, especially for big solid tumors. - Response: We have added to the following sentence, stating that “However, RIT is still a subject to radioresistance phenomenon, especially for big solid tumors and induces hematoxicity in some patients” .
(page 2, LN 53-54)
- L44 “it has low toxicity in comparison to other forms of chemotherapy”. Not true. RIT induces hematotoxicity which is one of the major issues impeding its use in clinic. Plus, a comparison between RIT and chemotherapy which are two different therapies with specific effects is a nonsense. - Response: Please see response to the previous question. (page 2, LN 53-54)
- L49 “In contrast to cancer when every cancerous cell has to be destroyed to prevent the resurgence of the disease […]. Not false but in the case of radiotherapy this is an approximation. RIT induces direct effects (towards the binding cells and the surrounding cells thanks to cross-fire and bystander effects) but induces also indirect abscopal effect activating immunity reaction leading to the death of non-irradiated cells. - Response: We have changed the sentence in question, it now reads: “In contrast to cancer when every cancerous cell has to be destroyed either directly by cross-fire and bystander effects or indirectly via abscopal effect to prevent the resurgence of the disease - low number of microbial cells remaining post RIT can be eliminated by the immune system”. (page 2, LN 66-69)
- The authors did not explain their choice of bismuth-213. One sentence or two in the introduction about this choice and the characteristics of alpha-emitter compared to other radionuclides would have been appreciated.- Response: We have now included the following sentence into the Introduction: “213Bi, an alpha-emitter with a short physical half life of 46 min, delivers its radiation impact within a short period of time, which can match the doubling time of the majority of the fungal pathogens. It gives 213Bi an advantage over other longer lived alpha emitters such as 225Actinium”. (page 2, LN 76-78).
- The authors should use the international units. In the case of radioactivity Curie (Ci) has been replaced by Becquerel (Bq) since several years now. Same for h (hours) instead of hr.- Response: We have changed the units throughout the revised manuscript.
- The authors often used the term “dose” to refer to the injected activity which is false. Internal dose only refers to dosimetry, i.e. the energy absorbed per unit of mass, expressed in Gray (Gy). The injected activity (counts) is not equal to the dose which can only be determined using dosimetry. The authors should replace the term “dose” by “injected activity” to avoid confusion for the reader. - Response: We have replaced the dose with the injected activity throughout the revised manuscript.
- Dosimetry study, which is probably the most important parameter concerning toxicity is missing. To conclude about the safety of the protocol and go to therapy experiments, an accurate intern dosimetry study is mandatory for each injected activity.- Response: The biodistribution data are acquired during Phase I clinical trials in human or veterinary patients by imaging patients at several time points post administration of a radiopharmaceutical, as well as collecting blood and urine and then performing dosimetry calculations, which give radiation doses to individual organs. However, our study was an analogue of a “first in man” study (here “first in dog”), which was a much smaller scale and collected basic safety information before initiation of a Phase I clinical trial in canine patients (companion dogs) with invasive fungal infections.
- Final molar activity (Am) of the radiolabeled mAb is missing in the Material & Methods section. - Response: We have now included the final molar activity of the radiolabeled mAb which was 5 kBq/pmol. (page 9, LN 253)
Reviewer 2 Report
The submission is a safety study of Bi-213-labeled antibody 400-2 in healthy dogs, showing only mild toxicity. The authors conclude radiolabeled 400-2 has the potential as adjuvant therapy for invasive fungal infections. Nuclear therapy with radiolabeled antibodies is used only in oncology to date, while the authors propose a new application as infection therapy. I think that is a great concept, but I'm afraid the submission does not meet the scope of Molecules because the main research area is chemistry, but the submission has only a safety study. Anyway, I provided some specific comments below.
Major points
Study design. The authors conducted radiotoxic studies, but not biodistribution and dosimetry studies. In general, dosimetry is needed to evaluate the toxicity of radiopharmaceuticals. We cannot know the injected doses of around 4 mCi are appropriate or not, without dosimetry. The authors need to include dosimetry studies in dogs in the submission.
The first paragraph in the Results section. The administration route and dose are essential information because it is before the Materials and methods section.
- 70 and 95. The current submission does not include some data, but the authors don't need to hesitate to show data because the journal has no restriction on the length.
- 178-182. The authors should provide evidence to show the low cross-activity of 400-2 to human and canine before clinical studies.
Administration of radiolabeled antibody to dogs and follow up analysis. There are only radioactive doses but no protein ones. The route of administration is required.
Minor points
The title and l. 55. The author may want to change 213Bismuth to Bismuth-213.
L. 164. What is Hp(10) of 6 micro Sv Hp(10) per dog?
Author Response
Reviewer 2
The submission is a safety study of Bi-213-labeled antibody 400-2 in healthy dogs, showing only mild toxicity. The authors conclude radiolabeled 400-2 has the potential as adjuvant therapy for invasive fungal infections. Nuclear therapy with radiolabeled antibodies is used only in oncology to date, while the authors propose a new application as infection therapy. I think that is a great concept, but I'm afraid the submission does not meet the scope of Molecules because the main research area is chemistry, but the submission has only a safety study. Anyway, I provided some specific comments below. - Response: We would like to thank the Reviewer for his/her encouraging opinion about the novelty of our work. In regard to the suitability of this manuscript for Molecules, it was invited for the Special Issue devoted to metal-based radiopharmaceuticals and we think that an antibody radiolabeled with 213Bismuth for treatment of fungal infections falls under the scope of that Special Issue.
Major points
Study design. The authors conducted radiotoxic studies, but not biodistribution and dosimetry studies. In general, dosimetry is needed to evaluate the toxicity of radiopharmaceuticals. We cannot know the injected doses of around 4 mCi are appropriate or not, without dosimetry. The authors need to include dosimetry studies in dogs in the submission. - Response: The biodistribution data are acquired during Phase I clinical trials in human or veterinary patients by imaging patients at several time points post administration of a radiopharmaceutical, as well as collecting blood and urine and then performing dosimetry calculations, which give radiation doses to individual organs. However, our study was an analogue of a “first in man” study (here “first in dog”), which was a much smaller scale and collected basic safety information before initiation of a Phase I clinical trial in canine patients (companion dogs) with invasive fungal infections.
The first paragraph in the Results section. The administration route and dose are essential information because it is before the Materials and methods section. - Response: We have included the following sentence into the Results section: “On separate days, three dogs were administered intravenously 155.3, 142.5 or 133.2 MBq (1→3)‐β‐glucans targeting antibody 400-2 labeled with 213Bi, respectively”. (page 3, LN 94-95)
70 and 95 The current submission does not include some data, but the authors don’t need to hesitate to show data because the journal has not restriction on the length. - Response: We have included two additional figures into the revised manuscript - Figure 3 shows MCV values (page 4-5, LN 130-134), and Figure 5 - chloride, phosphorus and potassium values (page 6. LN159-162).
178-182. The authors should provide evidence to show the low cross-activity of 400-2 to human and canine before clinical studies. - Response: The low toxicity of 213Bi-labeled 400-2 antibody is due to its low cross reactivity with canine tissues, otherwise significant toxicity would have been observed. We will perform the evaluation of potential cross reactivity of 400-2 antibody with all major human tissues before initiating a Phase I clinical trial in human patients, which a regulatory requirement by both Health Canada and the US Food and Drug Administration.
Administration of radiolabeled antibody to dogs and follow up analysis. There are only radioactive doses but no protein ones. The route of administration is required. - Response: We have included the intravenous route of administration and the specific activity of the radiolabeled antibodies (370 kBq/µg) into the revised manuscript. (page 10, LN 276-277).
Minor points
The title and l. 55. The author may want to change 213Bismuth to Bismuth-213. - Response: We have changed 213Bismuth to Bismuth-213 in the revised manuscript.
- 164 What is Hp(10) of 6 micro Sv Hp(10) per dog? - Response: We have expanded the sentence in the revised manuscript, it now reads: “In regard to the radiation safety, the research and veterinary staff administering the 213Bi-labeled antibody and providing animal care received a combined dose of no more than 6 µSv personal dose equivalent at the depth of 10 mm (Hp(10)) per handled dog as measured by electronic dosimeters due to the 440 keV gamma emissions from the 213Bi”. (page 8, LN 214-215).
Reviewer 3 Report
This is an important articel worthy of publication. It is a step towards application of labeled antibody to (1→3)‐β‐glucan to therapy of invasive fungal infections. In this work authors tested the toxicity of 213 Bi labeled antibody. Blood and urine various parameters of 3 dogs after injection of the drug were tested over a long period of time.
The publication is well written and I did not find any significant errors. There is only one important issue that the authors should address. As is known, monoclonal antibodies (proteins) have very slow pharmacokinetics in humans (about 1-3 days) and labeled 213Bi (T12 = 46 min) will not allow effective therapy. As far as I know, 213Bi is not used to label whole antibodies. Thus, first of all, before the toxicity tests are studied, therapeutic efficacy studies must be performed. Such studies were conducted by the authors, but on a very small mouse model, where, however, the pharmacokinetics are much faster (Front. Microbiol. 11:147). Therefore, I would ask the authors to respond to this issue and add the appropriate comment in the publication.
Author Response
Reviewer 3
This is an important articel worthy of publication. It is a step towards application of labeled antibody to (1→3)‐β‐glucan to therapy of invasive fungal infections. In this work authors tested the toxicity of 213 Bi labeled antibody. Blood and urine various parameters of 3 dogs after injection of the drug were tested over a long period of time. The publication is well written and I did not find any significant errors. There is only one important issue that the authors should address. As is known, monoclonal antibodies (proteins) have very slow pharmacokinetics in humans (about 1-3 days) and labeled 213Bi (T12 = 46 min) will not allow effective therapy. As far as I know, 213Bi is not used to label whole antibodies. Thus, first of all, before the toxicity tests are studied, therapeutic efficacy studies must be performed. Such studies were conducted by the authors, but on a very small mouse model, where, however, the pharmacokinetics are much faster (Front. Microbiol. 11:147). Therefore, I would ask the authors to respond to this issue and add the appropriate comment in the publication. - Response: We would like to thank the Reviewer for his/her encouraging opinion about our work. We have included the following sentence in the Discussion: “213Bi-labeled whole antibodies have been successfully used in treatment of patients with acute myeloid leukemia and melanoma (reviewed in (3)). Currently there is a world wide effort to increase the availability of 213Bi which will enable the expansion of 213Bi-labeled whole antibodies use for the variety of oncological and infectious disease applications.” (page 9. LN 233-237).
Round 2
Reviewer 1 Report
The authors have revised all the issues. The manuscrit is now suitable for publication in Molecules.
Reviewer 2 Report
The authors provided appropriate responses to all the comments, so the current submission meets the criteria to be published. However, I think the authors should have provided dosimetry studies in dogs to let readers know the absorbed dose in the present study.